# Identifying Drivers of Predictive Uncertainty using Variance Feature Attribution

## Abstract

Explainability and uncertainty quantification are two pillars of trustable artificial intelligence. However, the reasoning behind uncertainty estimates is generally left unexplained. Identifying the drivers of uncertainty complements explanations of point predictions in recognizing potential biases and model limitations. It additionally facilitates the detection of oversimplification in the uncertainty estimation process. Explanations of uncertainty enhance communication of and trust in decisions. They allow for verifying whether the main drivers of model uncertainty are relevant and may impact model usage in certain applications. So far, the subject of explaining uncertainties has been rarely studied. The few exceptions in existing literature are tailored to Bayesian neural networks or rely heavily on technically intricate approaches, such as auxiliary generative models, thereby hindering their broad adoption. We propose variance feature attribution, a simple and scalable solution to explain predictive aleatory uncertainties. First, we estimate uncertainty as predictive variance by adapting a neural network, for example, by equipping it with a Gaussian output distribution. We achieve this by adding a variance output neuron and can thereby rely on pre-trained point prediction models and fine-tune them for meaningful variance estimation. Second, we apply out-of-the-box explainers on the variance output of these models to explain the uncertainty estimation. This two-step method can be easily applied to any neural network with model-agnostic or model-specific explainers. We evaluate our approach in a synthetic setting where the data-generating process is known. We show that our method can explain uncertainty influences more reliably and faster than the established literature baseline CLUE, while the uncertainty estimation stage does not impede the accuracy of the model. As an illustrative application, we fine-tune a state-of-the-art age regression model to estimate uncertainty and generate attributions for age prediction uncertainty. Our exemplary explanations highlight reasonable potential sources of uncertainty, such as laugh lines and frowning. Variance feature attribution provides accurate explanations for uncertainty estimates with little modifications to the model architecture and low computational overhead.

## 1 Introduction

Researchers have recognized the importance of uncertainty quantification and explainability of machine learning (ML) predictions to ensure the successful real-world adoption of ML-based systems in safety-critical applications (Abdar et al., 2021; Vilone & Longo, 2020). These dimensions serve as key indicators of a model's trustworthiness, reliability, and fairness, which are crucial for its broad acceptance and use (Lambert et al., 2022; Lötsch et al., 2022). Predictive uncertainty in ML refers to the degree of confidence associated with a model's predictions (Chua et al., 2023). It can be decomposed into an epistemic and aleatory component (Kendall & Gal, 2017). Epistemic uncertainty arises from the scarcity of data in specific areas of the input space, for example, because a particular condition may be underrepresented. Principally, it can be reduced by acquiring additional examples of this condition. Aleatory uncertainty refers to inherent randomness or variability in the data, representing uncertainty that cannot be reduced by including additional training examples. Such uncertainty can arise due to measurement errors or certain variables relevant to the observed process not being collected. Uncertainty estimation is critical in risk management. It allows taking

conservative action, relying on the model only when it exhibits a high degree of confidence in its predictions, and avoiding usage outside its area of competence (Kompa et al., 2021).

Explainability encompasses methods that enhance the transparency of ML models by highlighting how features influence the model's output or by rendering the internal computations of black-box models more interpretable. Explainability methods enable understanding whether a model has learned relevant patterns from the input data and can reveal interesting previously unknown associations (Samek et al., 2021; Schwalbe & Finzel, 2023). Uncertainty quantification and explainability ensure accountable, informed and, therefore, responsible decision making and help mitigate biases and risks (Bhatt et al., 2021; Zhou et al., 2022; McGrath et al., 2023).

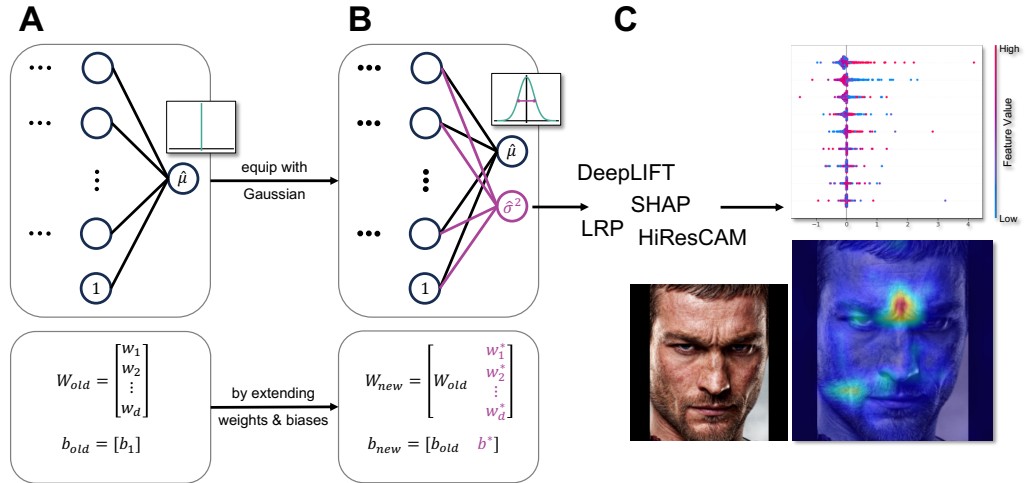

Figure 1: Overview of the variance feature attribution pipeline. (A) An original point prediction model with an output layer with weight matrix $W \in \mathbb{R}^{d \times 1}$ and a scalar bias. We equip this model with a Gaussian distribution resulting in (B), a model with output weight matrix $W \in \mathbb{R}^{d \times 2}$ and bias $b \in \mathbb{R}^2$. The two outputs are the mean $\hat{\mu}$ and the variance parameters $\hat{\sigma}^2$ of the predictive distribution. (C) From there, we can explain the variance using any suitable explainability method resulting in input-specific feature attributions that can be used to understand the drivers of the model's uncertainty.

In the majority of applications, the focus of model explainability predominantly lies in interpreting point predictions (Vilone & Longo, 2020). There is a significant gap in understanding and explaining the drivers of uncertainty estimates. In practice, when a machine learning algorithm is deployed and yields a substantial uncertainty estimate for a specific instance, the possible courses of action involve abstaining from employing the model if alternatives are available or accepting an increased level of risk. With explainable uncertainties, however, users gain the capability to identify the factors contributing to elevated uncertainty levels. This understanding allows domain experts to judge their relevance in a given scenario. Additionally, it provides valuable insights concerning necessary modifications to augment the model's predictive certainty and performance. In cases where abstaining from model usage is still necessary, factors influencing the decision can be understood and communicated. For example, if such an uncertainty factor is a feature indicating a person's minority status, it could suggest a potential bias of the model. The bias would be undetectable by naive explanations if the feature only influences the uncertainty estimation but not the mean prediction.

Explanations can be categorized as either local or global (Schwalbe & Finzel, 2023; Adadi & Berrada, 2018). Local explanations provide insights into how a model makes predictions for a specific instance. A local explanation of the model's uncertainty could foster more transparent discussions about ML-assisted decisions and risks, leading to increased trust. Global explanations provide an overview of a model's behavior across the entire input space. In the realm of uncertainty explanations, they can be used to detect general drivers of uncertainty and certainty or to partition the input space into easy- vs. hard-to-predict regions. This knowledge can be utilized to formulate hypotheses to improve the model or to detect unintended shortcuts in the uncertainty estimation process, such as spurious correlations or biases.

There is little prior work on explaining uncertainties, and existing literature mainly focuses on classification and generally relies on Bayesian neural networks (BNNs) or technical intricacies such as auxiliary generative models (Antoran et al., 2021; Perez et al., 2022; Ley et al., 2022; Wang et al., 2023). BNNs assign probability distributions to network weights to capture uncertainty (MacKay, 1992). However, due to their computational complexity and altered training process, BNNs have not been as widely adopted as classical neural networks. This hinders the adoption of BNN uncertainty explanation approaches (Lakshminarayanan et al., 2017).

We propose a simple and scalable solution for explaining uncertainties that can be readily integrated into existing ML pipelines (see Figure 1). We propose extending existing pre-trained point prediction models to additionally estimate parameters of the spread of a given probability distribution. In this work, we predict parameters of a Gaussian distribution similar to a classic heteroscedastic regression model. We approximate the error distribution by a Gaussian as its variance parameter can be interpreted as a measure of the aleatory uncertainty of the model. We can then use any state-of-the-art explainability method to explain the variance estimate provided by this probabilistic model. By highlighting the features contributing to the variance output, we can identify the input features that contribute to the model's uncertainty. While we focus on heteroscedastic regression in this work, the approach can be generalized to other target spaces, for example, for classification. To evaluate our method, we construct a synthetic problem where we can control how data is generated and how features contribute to the aleatory uncertainty present in the data. We train a simple feed-forward neural network on the regression task and test how many injected uncertainty-driving features we can recover. We compare our method to the established literature baseline Counterfactual Latent Uncertainty Explanations (CLUE) (Antoran et al., 2021). Furthermore, we illustrate a potential application of our method to an age regression task on images. We extend a pre-trained state-of-the-art transformer model and analyze the feature-attribution maps obtained for the uncertainty estimate.

## 1.1 RELATED WORK

Uncertainty quantification and explainability are rich areas of research (Abdar et al., 2021; Vilone & Longo, 2020). Yet, few researchers have recognized the importance of explaining the sources of uncertainty in predictions. Yang & Li (2023) have developed an explainable uncertainty quantification approach for the prediction of molecular properties. They employ message-passing neural networks and generate unique uncertainty distributions for each atom of a molecule. However, this approach is inherently specialized for graph-based representations of molecules. Counterfactual Latent Uncertainty Explanations (CLUE) (Antoran et al., 2021) and related approaches (Perez et al., 2022; Ley et al., 2022) derive counterfactual explanations by optimizing for an adversarial input that is close to the original input but minimizes uncertainty. The adversarial input is constrained to the data manifold with a deep generative model of the input data to prevent out-of-distribution explanations. This necessitates an optimization process for each explanation and the training of an auxiliary generative model, rendering CLUE and its extensions computationally demanding and difficult to implement. While CLUE is applicable to other probabilistic ML methods, the authors focus on explaining BNNs. Wang et al. (2023) have developed a gradient-based uncertainty attribution method for image classification with BNNs. They modify the backpropagation to attain complete, non-negative pixel attribution and to prevent vanishing gradient issues. To develop a method for detecting deterioration in model performance and explaining its cause, Mougan & Nielsen (2023) use traditional ML methods and bootstrapping to obtain estimates of epistemic uncertainty. To explain the sources of uncertainty, they train a model and obtain uncertainty estimates on a test set transformed with an artificial distribution shift. In a second step, they train another model that predicts the uncertainty estimates obtained in the first step. Subsequently, Shapley values are estimated for the second model to extract the feature attributions for the uncertainty. Mehdiyev et al. (2023) also focus on traditional ML methods and employ quantile regression forests in operations research to obtain prediction intervals that quantify uncertainty. They extract feature attributions for the uncertainty by estimating Shapley values directly for these prediction intervals as output. Watson et al. (2023) take an information-theoretic approach and explain uncertainty by extending Shapley values to the predictive distribution's higher-order moments in a conditional entropy setting for classification.

## 2 METHODS

### 2.1 DEEP HETEROSCEDASTIC REGRESSION

We want to explain uncertainties in the context of deep distributional neural networks that go beyond mere point predictions, i.e., they estimate the parameters of a specified output distribution. For this work, we focus on deep heteroscedastic regression with a Gaussian output, where we capture the mean and variance of the target, thereby directly modeling input dependence of the output noise. Here, we consider a regression setting with $n$ independent training examples $\{(\boldsymbol{x}_i, y_i)\}_{i=1}^n$ with input feature vector $\boldsymbol{x}_i \in \mathbb{R}^k$ and targets $y_i \in \mathbb{R}$, $i = 1, \ldots, n$. Instead of providing a full picture of the conditional distribution of the target, deep regression models usually only estimate its conditional mean by optimizing the mean squared error (MSE) or comparable loss functions. As a result, uncertainty information, which can be essential for decision-making and risk assessment, is not captured (Li et al., 2021). In contrast, we assume a heteroscedastic Gaussian as the conditional distribution

$$y \mid \boldsymbol{x} \sim \mathcal{N}\left(\mu_{\boldsymbol{x}}, \sigma_{\boldsymbol{x}}^2\right) \tag{1}$$

and represent its mean $\mu_{\boldsymbol{x}}$ and variance $\sigma_{\boldsymbol{x}}^2$ using a neural network $f_\theta : \mathbb{R}^k \to \mathbb{R} \times \mathbb{R}^+$ with weights $\boldsymbol{\theta}$. $k$ is the dimension of the input, and two output neurons produce the mean and variance estimates

$$f_{\boldsymbol{\theta}}\left(\boldsymbol{x}\right) = (\hat{\mu}_{\boldsymbol{x}}, \hat{\sigma}_{\boldsymbol{x}}^2), \tag{2}$$

respectively. We can then optimize the Gaussian negative log-likelihood

$$\mathcal{L} \propto \sum_{i=1}^n \left( \log(\hat{\sigma}_{\boldsymbol{x}_i}^2) + \frac{(y_i - \hat{\mu}_{\boldsymbol{x}_i})^2}{\hat{\sigma}_{\boldsymbol{x}_i}^2} \right) \tag{3}$$

and interpret the predicted variance as a measure of the aleatory uncertainty of the model. However, naively optimizing this criterion with overparametrized models such as deep neural networks can be unstable. These models frequently overfit by excessively shrinking the variance estimate or underfit by predicting only a mean estimate of the target and fitting the variance to the overall target variation in the data (Kuprashevich & Tolstykh, 2023; Wong-Toi et al., 2023; Nix & Weigend, 1994; Seitzer et al., 2022). In practice, these convergence difficulties can be mitigated by initially training the model using solely the MSE $\sum_{i=1}^n (y_i - \hat{\mu}_{\boldsymbol{x}_i})^2$ and subsequently switching to the Gaussian loss as in Equation (3) (Sluijterman et al., 2023). Additionally, instead of applying separate networks, mean and variance are estimated with shared hidden representations (Stirn et al., 2023).

### 2.2 EXTENDING PRE-TRAINED MODELS TO OUTPUT UNCERTAINTY ESTIMATIONS

The two-stage training process with mean warmup, as described above, naturally fits into the framework of transfer learning: The MSE-based initial training can be seen as a pre-training phase. The model is subsequently fine-tuned, switching to the Gaussian negative log-likelihood loss to capture predictive uncertainty (Equation (3)). The variance estimate is constrained to be positive with an exponential transformation and trained alongside the mean estimate using the Gaussian negative log-likelihood. In a scenario where pre-training with the MSE-loss and fine-tuning with the Gaussian negative log-likelihood is conducted jointly, it is possible to directly construct the neural network with the additional output neuron and exclude it when optimizing the MSE. However, in practice, it might be interesting to extend existing pre-trained models to capture uncertainty, where, for example, the model size and associated training costs make full re-training unfeasible. In this case, we can readily extend such pre-trained regression models by concatenating a column of randomly initialized weights to the weight matrix of the output layer to attain a variance estimate, i.e., an additional output neuron. We define the parameter matrix connecting the activations of the last hidden layer to the output as:

$$\boldsymbol{W} = \begin{bmatrix} w_1 \\ \vdots \\ w_d \end{bmatrix}. \tag{4}$$

Therefore, the final scalar output of a point prediction neural network can be calculated as:

$$\boldsymbol{x}\boldsymbol{W} + b = \hat{y} = \hat{\mu}_{\boldsymbol{x}}, \tag{5}$$

where $\boldsymbol{x} = [x_1 \quad \cdots \quad x_d]$ is the activation vector of the last hidden layer and $b$ is the output layer's bias term. To adjust the network to produce both parameters of a Gaussian distribution $(\mu_{\boldsymbol{x}}, \sigma_{\boldsymbol{x}}^2)$ and not just a point estimate $(\mu_{\boldsymbol{x}})$, we inject a parameter column in the last layer's weight matrix that will result in an additional output being produced by the neural network:

$$\boldsymbol{x} \times \begin{bmatrix} & w_1^* \\ \boldsymbol{W} & \vdots \\ & w_d^* \end{bmatrix} + \begin{bmatrix} b & b^* \end{bmatrix} = \begin{bmatrix} \hat{y}_1 & \hat{y}_2 \end{bmatrix} = \begin{bmatrix} \hat{\mu}_{\boldsymbol{x}} & \hat{\sigma}_{\boldsymbol{x}}^2 \end{bmatrix}, \tag{6}$$

where $\boldsymbol{w}^*$ is the randomly initialized new weight vector. Additionally, the bias term needs to be extended by $b^*$ to account for the increased number of output neurons. Alternatively, in architectures with a dedicated multi-layer regression head, it is equally straightforward to achieve an additional variance output neuron by adding a completely separate multi-layer variance head. We enforce the positivity of the variance with a suitable activation, such as the exponential function.

## 2.3 Post-hoc Explanation of Predictive Variance

Classical explainability methods focus on explaining the predicted class in a classification scenario or the point prediction in a regression task. We similarly explain the variance output in a heteroscedastic regression model. In these models, variance is an additional output to which we can apply any existing, appropriate explainability method. Variance feature attribution should not be understood as a new feature attribution method but as an extension of the scope on which such methods are applied. The choice of a suitable method follows the same criteria as for the explanation of the mean prediction or point predictions, and relevant literature should be consulted (Vilone & Longo, 2020; Samek et al., 2021; Schwalbe & Finzel, 2023). In principle, we can formulate our uncertainty explanation for any parametrized output distribution for which an explicit formulation of the uncertainty is available. In the case of a Gaussian output distribution, the application is most direct since the variance, as a parameter of the Gaussian, is an explicit output of the neural network. Furthermore, unlike distributions such as the Poisson or the exponential distribution, the variance is uncoupled from the mean output so that factors influencing the mean output are disentangled from uncertainty drivers.

In this work, we employ a model-agnostic and a model-specific explainability method to explain uncertainty. Model-specific methods are limited in the type of models that they can explain but may offer advantages such as lower computational complexity. In contrast, model-agnostic methods can be applied to any machine learning model (Adadi & Berrada, 2018). For our synthetic evaluation, we rely on KernelSHAP (Lundberg & Lee, 2017), a model-agnostic, local explainability method. KernelSHAP approximates Shapley values using a weighted linear surrogate model with an appropriate weighting kernel. For the age detection experiment, we use the CAM-based approach HiResCAM (Zhou et al., 2016; Draelos & Carin, 2021), a model-specific, local explainability technique that extracts an explanation by weighting the last feature map with the gradient of the output with respect to this last layer feature map. It was originally developed for convolutional neural networks and classification but can be applied to vision transformers and regression. We follow Chefer et al. (2021) who apply a similar CAM-based approach to the $[CLS]$ token of the last attention layer of the transformer model. Their approach can be readily adapted for transformers with class attention.

## 2.4 Synthetic Benchmark Pipeline

Evaluating explainability methods on real-world data is challenging due to the subjective nature of interpreting explanations based on expert prior knowledge. To address this, we employ synthetic data with a known data-generating process. Thereby, we can introduce controlled sources of heteroscedastic, aleatory uncertainty, which we subsequently aim to detect through our method, variance feature attribution. Specifically, we sample a synthetic ground truth using a linear system

$$\boldsymbol{\mu} = \boldsymbol{V}\boldsymbol{\beta} \tag{7}$$

with a design matrix $\boldsymbol{V} \in \mathbb{R}^{n \times p}$ with entries $\boldsymbol{V}_{ij} \sim \mathcal{N}(0,1)$, and ground truth coefficients $\boldsymbol{\beta} \in \mathbb{R}^p$ with $\boldsymbol{\beta}_i \overset{\text{iid}}{\sim} \text{Uniform}([-1,1])$. We introduce heteroscedastic noise sources with an absolute-value transformed polynomial model for the heteroscedastic noise standard deviation:

$$\boldsymbol{\sigma} = \mid \phi(\boldsymbol{U})\boldsymbol{\gamma} + \boldsymbol{\delta} \mid, \tag{8}$$

whereby $\boldsymbol{U} \in \mathbb{R}^{n \times p'}$ is a design matrix with entries $\boldsymbol{U}_{ij} \overset{\text{iid}}{\sim} \mathcal{N}(0, 1)$, $\phi(u_1, u_2, \ldots, u_{p'}) \rightarrow (u_1, \ldots, u_{p'}, u_1^2, u_1 u_2, \ldots, u_{p'}^2)$ is a second degree polynomial feature map, and $\boldsymbol{\delta} \sim \mathcal{N}\left(\boldsymbol{0}, \sigma_\delta^2 \boldsymbol{I}\right)$ is the uncertainty model error. $\boldsymbol{\gamma} \in \mathbb{R}^{\binom{p'+2}{2}-1}$ are ground truth noise coefficients with entries sampled from $\boldsymbol{\gamma}_i \sim \text{Uniform}([-1, -0.5] \cup [0.5, 1])$ to avoid negligible effects.

We can then sample the target $\boldsymbol{y} \in \mathbb{R}^n$ with

$$\boldsymbol{y} \sim \mathcal{N}\left(\boldsymbol{\mu}, \alpha \cdot \text{diag}(\boldsymbol{\sigma}^2) + \sigma_\epsilon^2 \boldsymbol{I}\right). \tag{9}$$

$\alpha \in \mathbb{R}^+$ determines the overall strength of the heteroscedastic uncertainty and $\sigma_\epsilon^2 \in \mathbb{R}^n$ regulates the homoscedastic noise.

For our experiments, we set $\alpha = 2.0$, $\sigma_\epsilon^2 = 0.02$, and $\sigma_\delta^2 = 0.05$ to get non-negligible, feature-dependent noise that can be modeled. We choose $p = 70$ and $p' = 5$ so that the uncertainty sources have to be detected among a larger set of features that do not influence the uncertainty. We sample $n = 41,500$ data points and concatenate both design matrices to attain the input $\boldsymbol{X}_{(n \times 75)} = \left[\boldsymbol{U}_{(n \times 5)}, \boldsymbol{V}_{(n \times 70)}\right]$ which we split into a train set, validation set, and test set of 32,000, 8,000, and 1,500 instances, respectively. We fit a deep neural network that has four hidden layers with 64, 64, 64, and 32 units. The network has two output neurons for the mean and variance prediction. During training, we apply dropout regularization with a dropout probability of 0.1 to the first two layers. We use the Adam optimizer and a batch size of 64. We commence the training using the MSE and subsequently optimize the model using the Gaussian negative log-likelihood as the loss function. This is akin to acquiring a pre-trained model and then refining it through fine-tuning. In both training phases, we stop training when improvement on the validation set ceases and choose the model weights that resulted in the smallest validation loss.

To interpret the uncertainties, we apply the post-hoc feature attribution method KernelSHAP from the SHAP package (Lundberg & Lee, 2017). We attain a feature importance measure as the mean absolute estimated Shapley values over the test dataset. We can then scrutinize if the noise sources $u_i$ are rediscovered as the most important features for the uncertainty estimation.

We compare our approach to Counterfactual Latent Uncertainty Explanations (CLUE), for which we have to train a variational autoencoder on the train data and apply the optimization as detailed in (Antoran et al., 2021). We calculate the CLUE feature importances as the mean of the absolute differences between each latent explanation to each original input over the test dataset.

## 2.5 Application to Age Detection

To showcase the applicability of our approach in a non-synthetic setting, we turn to the problem of age detection. Age detection is a relevant task in computer vision and finds application in a wide range of areas, from security to retail. We apply MiVOLO (Kuprashevich & Tolstykh, 2023), a recent state-of-the-art transformer-based model that achieves best-of-its-class performance in multiple age detection benchmarks (Kuprashevich & Tolstykh, 2023; Lin et al., 2022; Zhang et al., 2017). MiVOLO was designed explicitly for age detection, alongside the related problem of gender detection, and tackles both problems simultaneously to leverage synergies between the tasks. MiVOLO builds on VOLO (Yuan et al., 2023), a vision transformer architecture based on a special outlook attention that employs mechanisms usually used in convolutional neural networks. In contrast to VOLO, MiVOLO has two separate inputs. The first is the image patch with a person's face, and the second is the image patch of the body with the face removed. However, for simplicity, we use a version of the model that only uses the face input, bringing it closer to the original VOLO. The model produces a single three-dimensional output tensor, representing the prediction of the male and female sexes and the person's age.

We use a pre-trained version of MiVOLO and, following our procedure introduced in Section 2.2, extend the parameter matrices of the MiVOLO head, auxiliary head, and their respective bias terms. We initialize the extension of the parameter matrices using a Gaussian distribution following Glorot & Bengio (2010) and set the new bias terms to zero. The new dimension of the output vector is four, including the age prediction variance. After we have equipped the model with a Gaussian distribution, we train it using the IMBD-clean dataset (Lin et al., 2022), using the annotations and pre-processing by Kuprashevich & Tolstykh (2023). We fine-tune the model with the Gaussian

negative log-likelihood as our loss function. Consequently, we turn the previous point prediction for age into the mean of our estimated Gaussian distribution. The loss term for the gender detection remains a BinaryCrossEntropy loss. We use an Adam optimizer with a learning rate of $1e$-5, a weight decay of $1e$-2, and a batch size of 176. We use the validation set only for model selection. To detect and visualize the drivers of uncertainty in the images, we use HiResCAM as described in Section 2.3.

## 3 RESULTS

### 3.1 DETECTION OF NOISE SOURCES IN SYNTHETIC DATASETS

We examine the capability of our method to identify the drivers of uncertainty, which are features that correlate with the magnitude of the heteroscedastic noise. For the synthetic dataset, we know the data-generating process and, therefore, the ground truth noise sources and aim to rediscover them using variance feature attribution. An analysis of the quality of the uncertainty estimate can be found in the Appendix A.1. The results indicate that the uncertainty estimates generated by our model are meaningful; however, for them to be used to guide decision-making in practical applications, we argue that they should be explainable and ascertain if our method can detect the features influencing the heteroscedastic noise in the data. We use variance feature attribution with estimated Shapley values to delineate each feature's contribution to the model's uncertainty on 200 random test examples. The SHAP summary, depicted in Figure 2, offers an aggregated view of feature impacts, including their directionality. The five ground truth noise features are accurately identified as the top five contributors to model uncertainty (Figure 2 A).

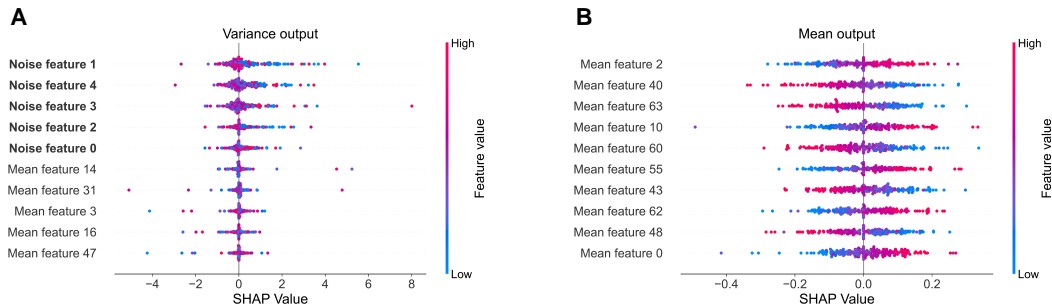

Figure 2: Explanations for uncertainty and mean predictions for the synthetic dataset. We display SHAP summaries for the 10 most important features of model uncertainty (A) or mean prediction (B) ordered by the mean of their absolute estimated Shapley values. We employ a neural network trained on synthetic data with heteroscedastic noise that non-linearly depends on 5 of 75 input features. By interpreting the variance output, we identify the key factors driving the aleatory uncertainty of the model. The attribution of the mean output offers complementary information but disregards uncertainty features.

In practice, explaining uncertainties is particularly relevant in instances with very high or very low uncertainty. In addition to random instances, we, therefore, apply variance feature attribution to the 200 highest and lowest uncertainty instances of the 1,500 test samples. We depict the resulting feature importance, measured as mean absolute estimated Shapley values, in Figure 3 A, C, E. For comparison, we explain the same instances for the same model using CLUE (Antoran et al., 2021) as an alternative explainer of model uncertainty (see Figure 3 B, D, F). We calculate CLUE feature importance as the mean absolute difference between CLUE uncertainty counterfactuals and the original sample. We find that variance feature attribution and CLUE both effectively identify uncertainty drivers for high-uncertainty instances. Variance feature attribution also delivers stable performance for random and low uncertainty examples while CLUE's detection capability deteriorates. This suggests that, unlike CLUE, variance feature attribution can elucidate the factors contributing to certainty in addition to those relevant to uncertainty. To compare the runtimes, we use an Intel(R) Core(TM) i5-12600K machine with an NVIDIA RTX 3060 Ti. We apply the post-hoc explanation of both methods to the same trained model 15 times. A run takes $81 \pm 34$ seconds for variance feature attribution and $1012 \pm 33$ seconds for CLUE. Notably, the uncertainty estimation training stage

only marginally reduces the accuracy of the regression model in our experiment. The MSE is 0.79 after the MSE pre-training stage and 0.80 after the uncertainty-aware fine-tuning stage. Naturally, the ability of variance feature attribution to detect drivers of uncertainty depends on the train dataset size and magnitude of the noise signal. We analyze this in Appendix A.2.

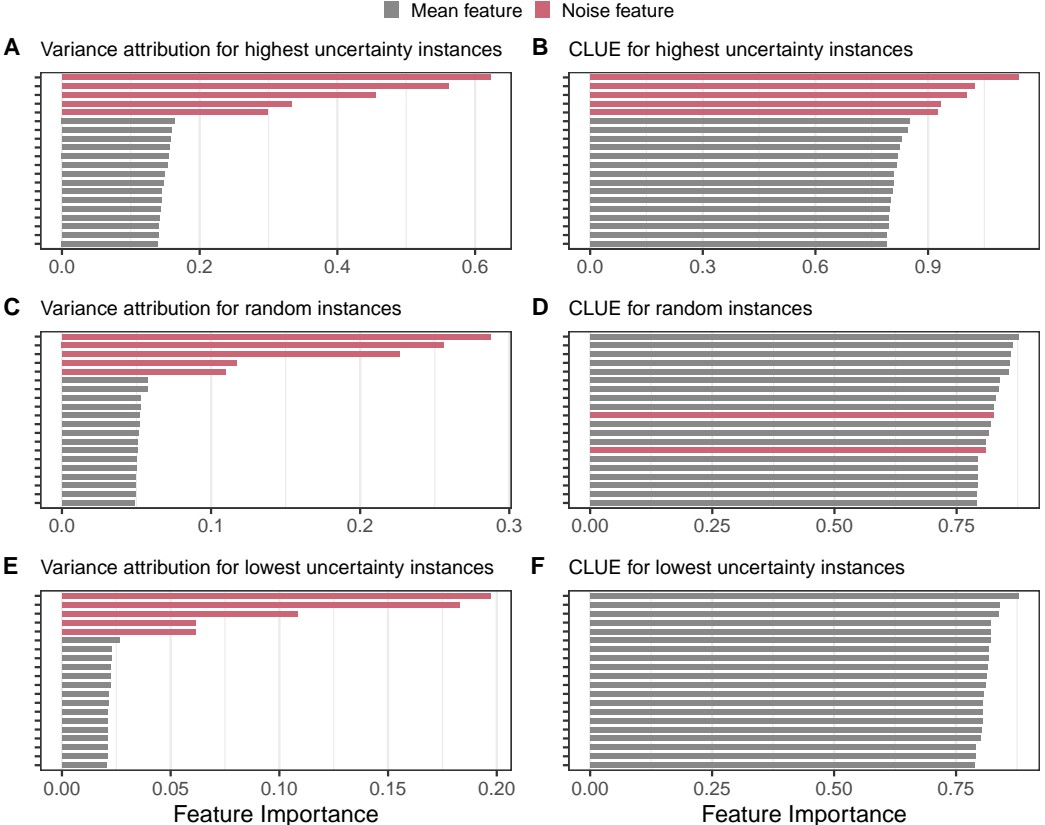

Figure 3: Top 20 variance feature attribution importances and CLUE feature importances on a synthetic dataset. A, B: From a test set of 1,500 samples with 70 mean and 5 noise features, we explain the 200 instances with the highest predicted uncertainty. Our variance attribution method (A) and CLUE (B) both faithfully assign high importance to the ground truth uncertainty sources. C, D: For 200 random instances, variance attribution delivers accurate explanations (C), while CLUE is unreliable (D). E, F: Variance feature attribution also can attribute (un-)certainty for the 200 lowest uncertainty predictions (E), while CLUE performance deteriorates (F).

## 3.2 Finding Potential Drivers of Uncertainty in Age Detection

We investigate the application of variance feature attribution to a non-synthetic age detection task using the MiVOLO model (Kuprashevich & Tolstykh, 2023) and the IMDB-clean dataset (Lin et al., 2022). Similar to our synthetic experiment, we also evaluate the quality of the uncertainty estimate, and the results can be found in the Appendix A.3. The results similarly suggest that the uncertainty estimates are relevant.

Applying our method to the MiVOLO model using HiResCAM as the explanation method reveals reasonable potential explanations for the predictive uncertainty (see Figure 4). The explanations mainly focus on areas around the eyes, mouth, nose, and forehead. These areas seem to be highlighted especially strongly when the person in the image shows emotions such as joy and anger that lead to distortions of these facial areas that the model might confuse with age-induced wrinkles. Similar highlights in the explanations are present throughout a majority of images in the test dataset.

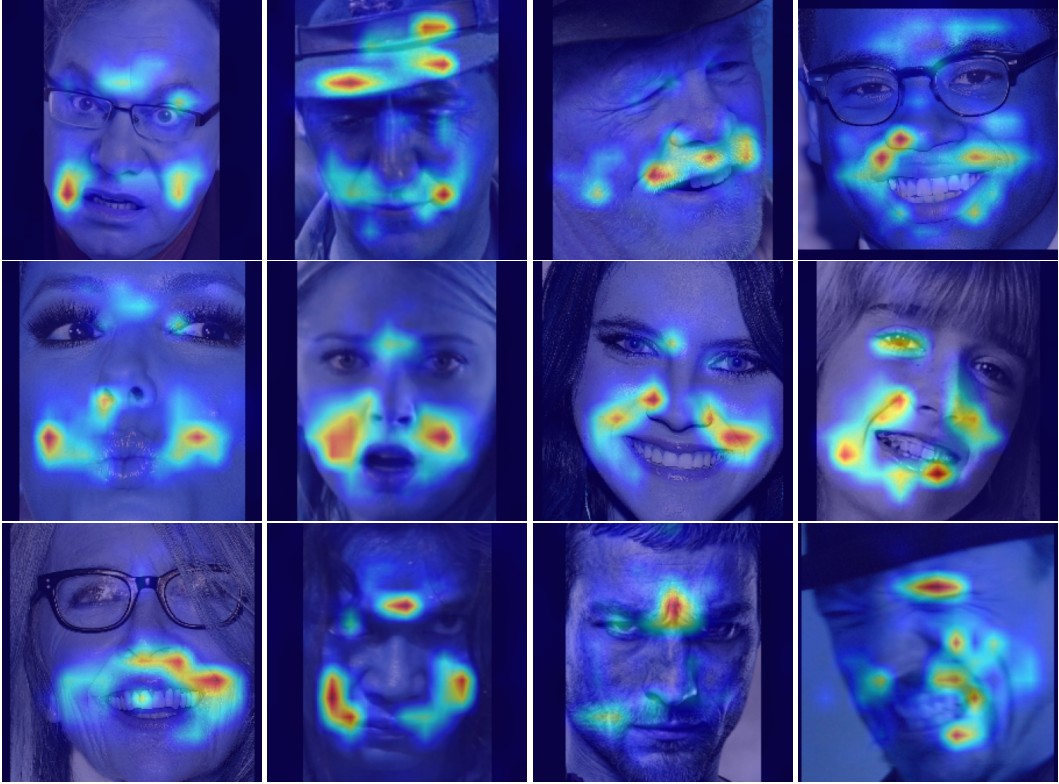

Figure 4: Examples of uncertainty explanations for images from MiVOLO's (Kuprashevich & Tolstykh, 2023) IMDB-clean-based (Lin et al., 2022) test set. Variance feature attribution using HiResCAM (Draelos & Carin, 2021) mainly highlights facial distortions such as laugh lines and frowning. The corresponding input images are depicted in Appendix A.3.

## 4 DISCUSSION AND CONCLUSION

In this work, we have proposed variance feature attribution, a simple strategy for explaining predictive aleatory uncertainties. This approach requires minimal modifications to existing neural network regressors, making it a practical solution for broad adoption. We use neural networks with Gaussian output distribution to estimate uncertainty and apply appropriate explanation methods to the variance output. Thereby, we are able to provide explanations for the sources of uncertainty estimates. In synthetic experiments, the resulting explanations can be generated faster while matching CLUE baseline performance or outperforming it in situations with low and medium uncertainty. Further, estimating the uncertainty does not considerably impede prediction accuracy. Generally, the Gaussian assumption may be unsuitable in some applications. In principle, we can extend this framework to predict parameters of various other distributions, such as more heavy-tailed or mixture distributions, providing the opportunity for explainable uncertainties tailored to specific problem domains. Since conventional evaluation metrics are not directly applicable, we have introduced an evaluation protocol designed to assess uncertainty explainers involving the use of synthetic data with a known ground-truth noise profile. We aim to expand this approach using more diverse data-generating processes or by fusing real data with synthetic noise sources. Generally, we inherit practical limitations of uncertainty estimation, such as overconfidence, as well as challenges related to explainability, such as issues with faithfulness and consistency. Nonetheless, significant advancements are occurring in both of these domains, which can be seamlessly integrated into our method. Future work might involve the study of synergies of the explanations of point and uncertainty predictions. For example, in the context of explainable active learning (Ghai et al., 2021), a shared visualization of both explainability modes could be beneficial.

REPRODUCIBILITY STATEMENT

Our research findings can be reproduced using the code and data provided in our anonymous git repository available at `https://anonymous.4open.science/r/vfa-variance-feature-attribution-0E63`, including scripts to create the figures and information on the requirements. For the synthetic experiment, this includes the pipeline that can be used to create additional datasets to probe the method further. Details on how to run the synthetic experiments and how to download the age detection data and reproduce our explanations are in the repository's README.

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

## A APPENDIX

### A.1 UNCERTAINTY EVALUATION IN THE SYNTHETIC EXPERIMENT

We assess the model's ability to learn meaningful uncertainties by adapting a calibration curve for regression uncertainties as described by Kuleshov et al. (2018). Precisely, we calculate the cumulative probability of the observations with respect to the predicted distribution

$$F_i = CDF_{Gaussian}\left(\frac{(y_i - \hat{\mu}(x_i))}{\sqrt{\hat{\sigma}^2(x_i)}}\right) \tag{10}$$

and calculate the empirical cumulative probability distribution at 20 probability levels $\tilde{p}_j = 0.05j; j = 1, 2, ..., 20$:

$$\hat{p}_j = \frac{|\{y_i \mid F_i < \tilde{p}_j, i = 0, 1, 2, ..., n\}|}{n}. \tag{11}$$

In Figure 5 (A), we plot the empirical frequency $\hat{p}_j$ against the probability levels $\tilde{p}_j$ and expect $\hat{p}_j \to \tilde{p}_j$ when $n \to \infty$ for perfectly calibrated models. The uncertainty estimates are overall well calibrated.

We further look at the reduction in test MSE we can achieve by limiting the test set to quantiles of the test set with the lowest uncertainty. This mimics a scenario where users would opt out of utilizing the prediction for high-uncertainty instances. As seen in Figure 5 (B), limiting the test set to low uncertainty predictions starkly reduces the test error. We compare this to a baseline where instances are included based on them having a low absolute distance from the mean prediction. The reduction is close to the theoretical optimum, where instances are removed based on the ground truth standard deviation of the heteroscedastic noise.

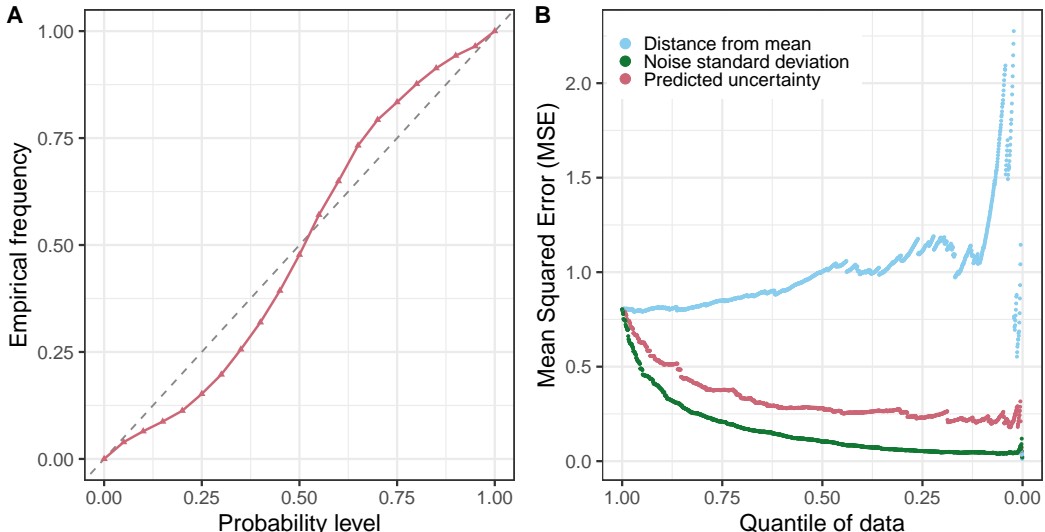

Figure 5: (A) Uncertainty calibration is illustrated by binning predictions into 20 probability level intervals and plotting the expected against the empirical frequency of observing the true target in each interval. The dashed line indicates theoretical uncertainties with perfect calibration (Kuleshov et al., 2018). (B) The MSE of the trained deep heteroscedastic regression model. The test set is iteratively limited to the quantile of the test data with the lowest predicted uncertainty (red), to instances with the lowest ground truth standard deviation of the heteroscedastic noise (green), or as a baseline based on instances with the lowest distance from the mean (blue).

### A.2 UNCERTAINTY DRIVER DETECTION: NOISE MAGNITUDE AND DATASET SIZE

Like in every supervised machine learning task, a model's ability to learn and detect the signal, which in this case is the noise characteristic, depends on the signal-to-noise ratio and the size of the available dataset. We investigate the precision of noise feature detection across varying dataset sizes and signal strengths. For uncertainty estimation, the scaling factor $\alpha$ of Equation (9) used for the heteroscedastic noiser serves as an indicator of signal strength. Fig. 6 displays the precision of the detection of heteroscedastic noise features for various dataset sizes and noise scalers. We compute the precision as the ratio of noise features that appear among the top five most important features. As the dataset size and the magnitude of the representable uncertainty grow, the capability to identify the origins of uncertainty improves.

### A.3 UNCERTAINTY EVALUATION AND INPUT IMAGES OF THE AGE DETECTION EXPERIMENT

We verify that our model has learned to estimate uncertainty via the variance of the predicted distribution. We use the same approach as in our synthetic experiment (see Appendix A.1) and iteratively remove the sample with the highest predicted variance from our testing set and compare the development of the MSE when removing samples based on their predicted age (see Figure 7). We show all exemplary explanations and their corresponding input in Figure 8.

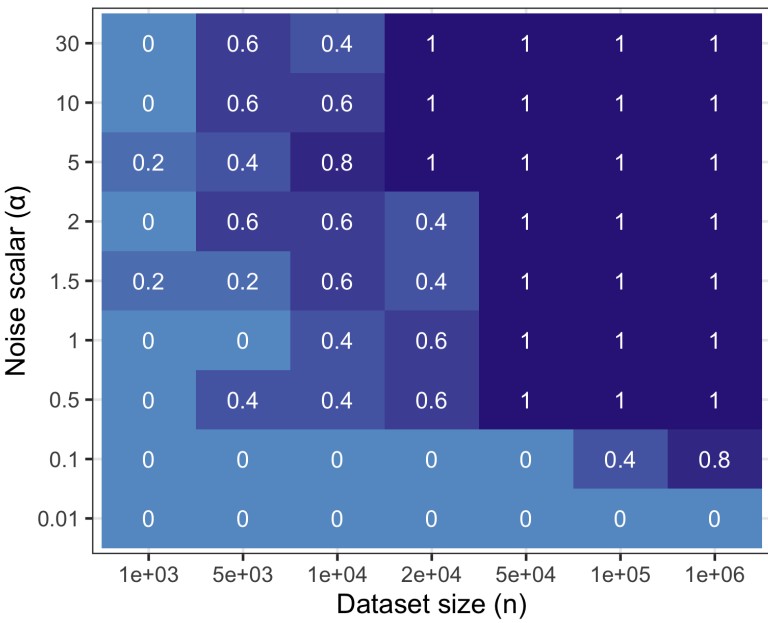

Figure 6: Precision of the detection of heteroscedastic uncertainty features vs. the dataset size and intensity of the heteroscedastic noise effect in the synthetic dataset. We use our proposed method in single trial runs. Small effects can be detected in large datasets.

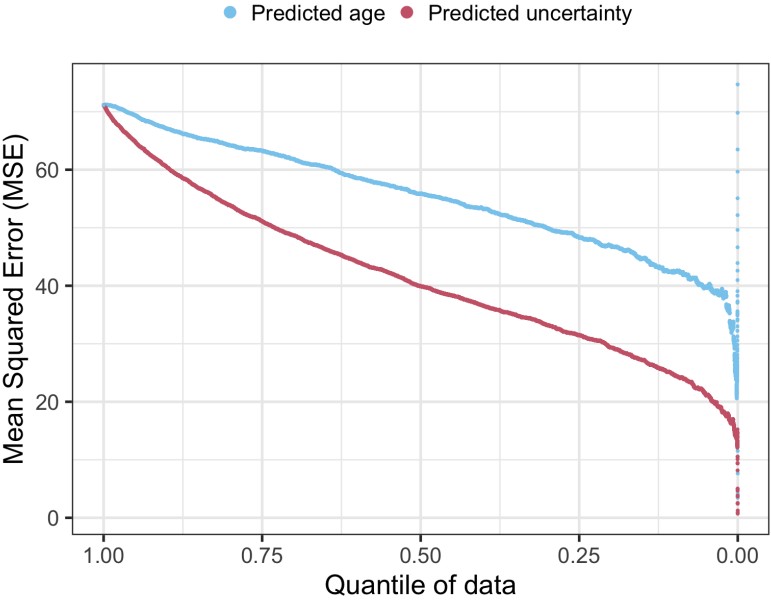

Figure 7: The test set MSE of the age detection MiVOLO model, fine-tuned for uncertainty estimation. The test set is iteratively limited to the quantile of the test data with the lowest predicted uncertainty (red). As a baseline, we remove instances according to the predicted age (blue), which consistently produces a set with higher MSE.

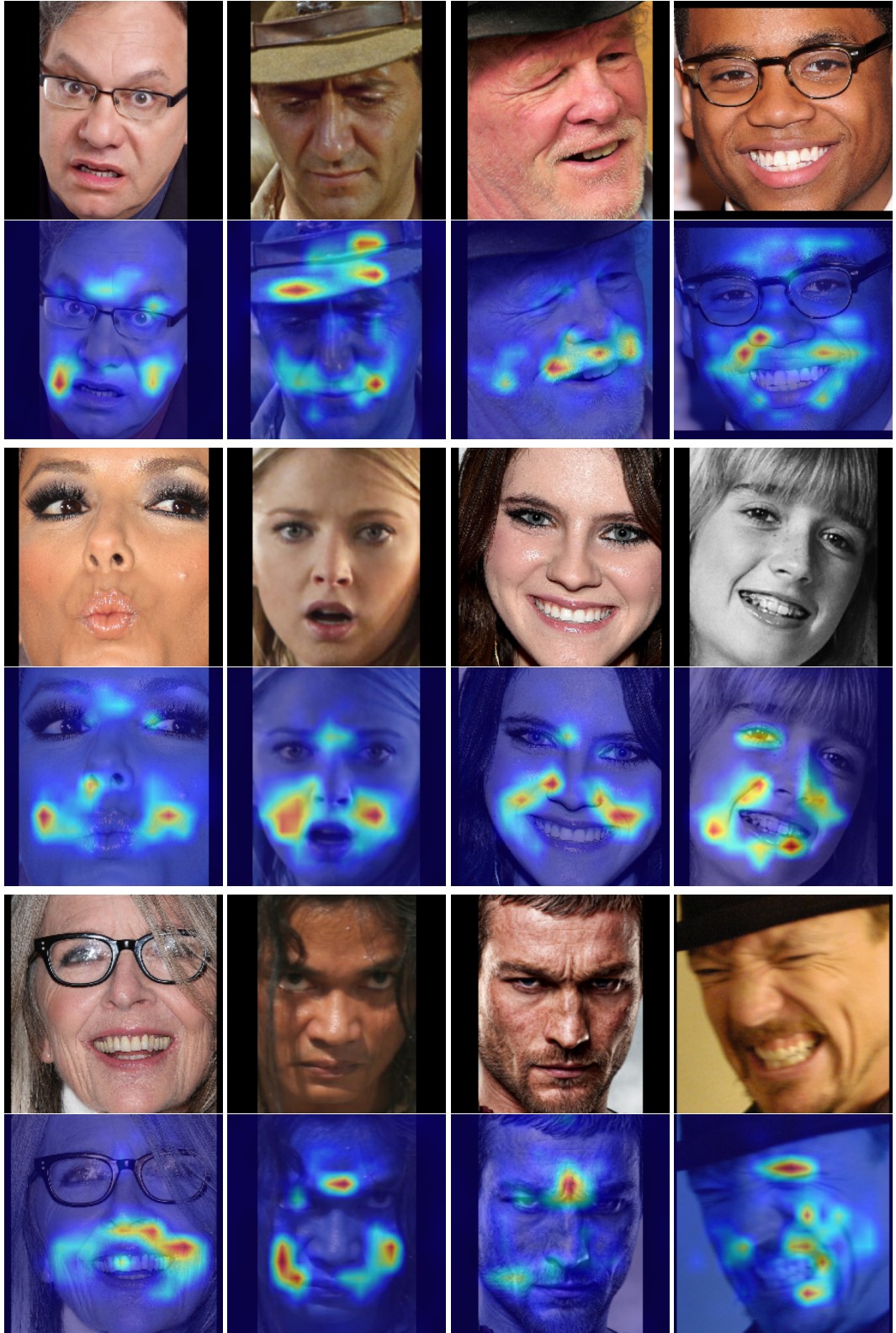

Figure 8: Input images and uncertainty explanations for images from MiVOLO's (Kuprashevich & Tolstykh, 2023) IMDB-clean-based (Lin et al., 2022) in our age detection experiment. Variance feature attribution using HiResCAM (Draelos & Carin, 2021) mainly highlights facial distortions such as laugh lines and frowning.

