# OpenReview forum: "Identifying Drivers of Predictive Uncertainty using Variance Feature Attribution"
_ICLR.cc/2024/Conference — ICLR 2024 Conference Withdrawn Submission_

### Official Review · Reviewer_3GEh · 2023-10-29

**Soundness:** 1 poor
**Presentation:** 3 good
**Contribution:** 1 poor
**Rating:** 1
**Confidence:** 5

**Summary:**

This paper is about explanations for aleatoric uncertainty of the variance output in a regression model trained with the Gaussian NLL. The authors propose to use gradient-based saliency maps to explain the variance head of a regression model under the Gaussian assumption. There are experiments on a synthetic dataset and on age regression.

Contributions are:
- A scalable solution to explain aleatoric uncertainty in a regression model.
- A method to extend pre-trained regression models to also consider aleatoric uncertainty, by training the variance head.
- A synthetic toy regression problem with controllable factors for heteroscedastic aleatoric uncertainty prediction, allowing for efficient evaluation.
- Results on the toy regression problem and age regression.

**Strengths:**

- The paper's writing is good and mostly easy to understand.
- The synthetic benchmark for evaluating aleatoric uncertainty explanations seems to be novel and significant (Sec 2.4). It basically uses Gaussian with a variable variance that has heteroscedastic and homoscedastic noise terms.

(Unfortunately I do not find more strengths in this paper)

**Weaknesses:**

- I believe that explaining aleatoric uncertainty is not a very interesting problem, while the authors argue about explaining uncertainty, aleatoric uncertainty is just the uncertainty in the data, usually noise in the labels, etc, and this does not have the same impact as epistemic (model) uncertainty, which is usually the kind of uncertainty that is interesting as it provides feedback about the prediction being correct or not.

- There are variations of the Gaussian NLL that have much less problems in optimization, like the beta-Gaussian NLL, etc. These variations are not used in this paper, which decreases the value to the community. Below I provide a reference to the beta-Gaussian NLL paper:

Seitzer, Maximilian, et al. "On the Pitfalls of Heteroscedastic Uncertainty Estimation with Probabilistic Neural Networks." International Conference on Learning Representations. 2021.

- Section 2.2 presents a method to add variance outputs to pr-etrained regression models, while I believe that this is a kind of trivial extension and there are no new ideas here, additionally the authors do not evaluate this proposed method, for example, some basic questions might arise, is it better to pre-train on MSE and later train on Gaussian NLL, or directly train on Gaussian NLL and one of its variations? It is not clear what is the value of this method as there is no evaluation or comparison.

- There is no proper comparison to the state of the art or ablation studies. The synthetic experiments are compared against CLUE, and it seems there is a small qualitative improvement (the paper does not use any quantitative metrics), but there is no comparison for the more interesting age regression experiment, which lowers the value of such comparison. Overall I understand that there is not much state of the art in this sub-field of explaining uncertainty, but then I suggest to perform ablation studies, compare multiple explanation methods, and multiple methods to estimate the variance of the data, to obtain useful insights for future research.

- I am not sure how to interpret the results of the age regression experiment (Figure 4). It is difficult to evaluate and interpret explanations, and I believe it is more difficult to explain uncertainties, the authors make qualitative comparisons among the explanations, which is fine, but how do these explanations relate to the aleatoric uncertainty? In the age regression example, aleatoric uncertainty labels are not available, so it is very difficult to argue that the model is explaining its aleatoric uncertainty. I believe this experiment requires more thinking and a proper experimental design, opposite of the synthetic experiment.

Minor Comments
- The paper refers to "Aleatory" uncertainty in some parts, but the actual technical name is aleatoric uncertainty.
- In Figure 4, only the saliency maps are presented, I believe there is more information to be presented, like the ground truth age, and predicted age mean and standard deviations, so the user can see how these three values relate to the saliency maps. Just by looking at the saliency maps without the predicted standard deviation is meaningless, as the whole aim of the paper is to explain the aleatoric uncertainty output head.

**Questions:**

- What is the interest of explaining aleatoric uncertainty, as opposed to explain epistemic uncertainty? I believe explanations of epistemic uncertainty are much more useful to end users, so what is the value of a aleatoric uncertainty explanation?

---

> ### Author Response · Authors · 2023-11-22
>
> Thank you for your thorough review of our submission, and we appreciate the time and effort you invested in providing detailed feedback.
> We acknowledge your perspective on the interest in explaining epistemic uncertainty as a more robust measure in out-of-distribution scenarios.
> We, however, think there is great value in explaining aleatoric uncertainty: We agree that aleatoric uncertainty is, as you mention, frequently noise in the labels. Mixture density networks model this noise conditioned on the input vector. We believe that it is crucial to know how a model came to the conclusion that a certain example has more noise in the label than another.
>
> Consider the example of an ML model predicting medical risk that has a higher uncertainty for instances from a certain demographic X even though demographic X is reasonably represented in the training data (as measured by: adding more data from demographic X only marginally decreases the uncertainty of the risk prediction).
> This aleatoric uncertainty will provide feedback about whether the prediction is correct or not (for demographic X, the model will be more likely to err). The explanation for the high aleatoric uncertainty would be very interesting (socio-economic status? genetic differences? higher median age?) and might lead to ideas on how to refine the model or which additional features to collect.
>
> Your suggestion regarding alternative variations of the Gaussian NLL, such as the beta-Gaussian NLL, is well-received. We previously implemented the beta-Gaussian NLL and ran experiments for our age regression showcase. However, we did not observe any improvements, which is why we forwent it. We will investigate this in more detail for the next versions.
> We also agree that it is difficult to interpret the results of the age regression experiment, and will provide additional information with the visualization as you proposed. We think it is generally difficult to evaluate explanation approaches without ground truth. That is why we devised the synthetic benchmark pipeline. Thank you for your suggestions in this area. We will devise a more quantitative evaluation strategy for this dataset that will strengthen our contribution.

---

### Official Review · Reviewer_H5dd · 2023-10-30

**Soundness:** 3 good
**Presentation:** 3 good
**Contribution:** 3 good
**Rating:** 6
**Confidence:** 4

**Summary:**

This paper proposes an XAI method based on the uncertainty of the prediction in the regression setting. The idea is to use a VAE-type of fitting with the black-box function $f(x)$ being the mean, assuming that all the training samples are available. Using the estimated predictive variance, the authors apply an existing attribution method of the Shapley value. In the computer vision domain, the authors report on interesting results.

**Strengths:**

- Uncertainty quantification in the context of XAI is a relatively new topic.
- The VAE-type variance estimation is a generic method and widely applicable as long as training data are available at hand.

**Weaknesses:**

- A few important contexts of the related work are missing.  A few recent works in the XAI community clearly point out the importance of uncertainty quantification. The following papers should be cited and discussed at least.
	-  Xingyu Zhao, Wei Huang, Xiaowei Huang, Valentin Robu, and David Flynn. 2021. BayLIME: Bayesian local interpretable model-agnostic explanations. In Proceeding of the 37th Conference on Uncertainty in Artificial Intelligence (UAI 21). PMLR, 887–896
	- Tsuyoshi Idé, Naoki Abe: Generative Perturbation Analysis for Probabilistic Black-Box Anomaly Attribution. KDD 2023: 845-856
- The method is data-hungry. The assumption of the availability of training data may be unrealistic.

**Questions:**

Please comment on the problem setting, where training data are assumed to be available, in light of prior works in the XAI research. Also, please clarify the novelty in light of the existing work, as pointed out above.

I will update my rating depending on your reply.

**Details Of Ethics Concerns:**

None.

---

> ### Author Response · Authors · 2023-11-22
>
> Thank you for your detailed review and valuable feedback on our submission. We appreciate the time and effort you've dedicated to evaluating our work. Here are our responses to your points:
> We appreciate your suggestion to include references to recent works in the XAI community and the field of uncertainty of explanations, specifically the papers by Zhao et al. and Idé and Abe. Zhao et al. propose a Bayesian extension to the LIME method with the purpose of including prior knowledge and improving consistency and robustness, which is, therefore, focusing on improving general explainability. Idé and Abe are concerned with the problem of anomaly attribution, where an observation strongly deviates from the prediction. Their proposed method allows for quantifying the uncertainty in the detected feature attributions while we explain the features affecting a model’s uncertainty. In future versions, we will make sure to position our work also in the field of uncertainty of explanations that is related to our research focus.
> Your concern about the assumption of the availability of training data is valid since we inherit every limitation of mixture density networks. We recognize that the assumption of sufficient training data may not always hold in practical scenarios.
> Thank you once again for your thoughtful review. We look forward to the opportunity to improve our work.

---

### Official Review · Reviewer_fhD9 · 2023-10-31

**Soundness:** 3 good
**Presentation:** 3 good
**Contribution:** 1 poor
**Rating:** 1
**Confidence:** 4

**Summary:**

The authors address the task of identifying drivers of predictive uncertainty. To this end, they follow a 2-step approach. First, they adapt a neural network to a mixture density network with an additional neuron capturing variance. Next, they compute KernelSHAP values on the uncertainty.

**Strengths:**

The approach addresses an interesting problem explaining the drivers of uncertainty; their approach is very simple and combines 2 well-known paradigms in a straight-froward manner.

**Weaknesses:**

- The main contribution of the paper is to demonstrate that the straight-forward combination of 2 well-known concepts (MDNs and KernelSHAP); for this to be a valuable resource I miss a  comparison to baselines (e.g. Watson et al as cited by the authors) and a systematic _quantitative_ evaluation on a representative number of real-world datasets. The qualitative evaluation on IMDB-clean is promising but does not warrant the strong conclusions of the authors
- Important literature missing: While the authors mention some  recent work in their discussion of deep heteroscedastic regression, they miss the large body of literature following the introduction of this very model in 1994 as Mixture Density Networks in Chris Bishop's seminal paper (which is not even cited); the generalisation to introduce the variance neuron after training a vanilla network is trivial

**Questions:**

See above

---

> ### Author Response · Authors · 2023-11-22
>
> Thank you for taking the time to review our manuscript. We highly appreciate your feedback and constructive comments. Below are our responses to your points:
> We understand your concerns about the perceived lack of a robust quantitative evaluation and a comparison to baselines. We plan to incorporate a systematic quantitative evaluation on a more representative set of real-world datasets and include comparisons with relevant baselines to strengthen our contribution for a resubmission.
> We acknowledge the oversight in not citing Chris Bishop's seminal paper on Mixture
> Density Networks from 1994 and will make sure to position our work more clearly concerning his and subsequent work in the field.
> We value your insights, and your feedback will significantly contribute to the enhancement of our work. We are committed to addressing these concerns and providing a more robust and well-supported contribution in future versions. As many SOTA explainability methods such as CLUE rely, e.g., on generative models and require significant compute resources and long runtimes, a comprehensive and fair comparison of several methods is not possible within the revision's timeframe.

---

### Official Review · Reviewer_Dpa4 · 2023-11-08

**Soundness:** 4 excellent
**Presentation:** 4 excellent
**Contribution:** 3 good
**Rating:** 8
**Confidence:** 3

**Summary:**

The paper focuses on the rarely studied subject of explaining uncertainties (versus explaining predictions). This area hasn't gotten a lot of attention because Bayesian approaches to neural networks hasn't been widely embraced (given the difficulties with training them). The authors propose "variance feature attribution" an approach to explain aleatoric uncertainty. They adapt a traditional neural network to be suitable by adding a variance output, and fine-tuning pre-trained point estimate models (the point becoming the mean of the distribution) to provide a useful variance estimation. They demonstrate their approach on a synthetic dataset such that the uncertainty can be controlled and compare their approach to explainability against CLUE. The goal is to identify which factors/features contribute to elevated (or reduced) levels of uncertainty. They also demonstrate their approach on a non-synthetic dataset (related to age regression) and show which areas of an image cause increase of uncertainty (marks around the eyes/mouth).

**Strengths:**

This is a well written paper that tackles an area that hasn't be given a lot of attention. I found the synthetic results very compelling, especially what was presented in Figure 3. I appreciate how they show that CLUE does explain the features causing uncertainty, but their approach makes the distinction between the features much more pronounced. The demonstration on a real-world dataset gives additional strength to their claims.

**Weaknesses:**

I had issues with Section 2.2, which I will discuss more in the questions. I believe some details were left out (or the authors thought they could be assumed) which would have made the section more explicit and clear. In that section the authors state "multi-layer regression head", which seems wrong to me. Is it suppose to be "multi-label regression head"? Also, once the additional output is added (to capture variance) and the Gaussian negative log-likelihood is used as a loss function, how many iterations should be performed?

**Questions:**

1. In section 2.2 do you mean "multi-label regression head" instead of "multi-layer regression head". If not, could you please elaboriate.
2. Can you explain why you used a batch size of 176 (it seems odd...well it is even, just uncommon).

---

> ### Author Response · Authors · 2023-11-22
>
> Thank you for your thorough review, insightful feedback, and for pointing out the strength of our approach to addressing the understudied problem of explaining uncertainty. We appreciate your feedback highlighting the need for additional detail in section 2.2. Regarding your comment on the “multi-layer regression head”: What we meant to convey is that certain models incorporate an MLP serving as a read-out function for a scalar output. In some cases, rather than merely extending the final layer, a second MLP can be introduced to generate the variance output. Consequently, this setup results in the presence of two “multi-layer regression heads”. We will ensure to clarify this aspect more explicitly in future iterations.
>
> Regarding the choice of a batch size of 176, this decision was made to optimize the GPU utilization within our local compute architecture. It was solely an engineering-based decision aimed at minimizing the training time.
>
> Due to the required time for performing extensions for other reviewer comments, we will not be able to submit a revised version to ICLR. Thanks again for your valuable input that will help improve the presentation of our work.